# Stratum Corneum Lipids in Non-Lesional Atopic and Healthy Skin following Moisturizer Application: A Randomized Clinical Experiment

**DOI:** 10.3390/life14030345

**Published:** 2024-03-06

**Authors:** Malin Glindvad Ahlström, Rie Dybboe Bjerre, Magnus Glindvad Ahlström, Lone Skov, Jeanne Duus Johansen

**Affiliations:** 1National Allergy Research Centre, Department of Dermatology and Allergy, Herlev and Gentofte Hospital, University of Copenhagen, DK-2900 Hellerup, Denmark; riedybboeolsen@gmail.com (R.D.B.); jeanne.duus.johansen@regionh.dk (J.D.J.); 2Department of Virus and Microbiological Special Diagnostics, Statens Serum Institute, DK-2300 Copenhagen, Denmark; magnus.rasch@gmail.com; 3Department of Dermatology and Allergy, Copenhagen University Hospital—Herlev and Gentofte, DK-2900 Herlev, Denmark; 4Department of Clinical Medicine, Faculty of Health and Medical Sciences, University of Copenhagen, DK-2900 Hellerup, Denmark

**Keywords:** atopic dermatitis, ceramides, moisturizer, lipid, skin barrier, stratum corneum

## Abstract

Introduction: It is an international standard to recommend patients with atopic dermatitis (AD) to use moisturizers; however, little is known about their effect on lipids in the stratum corneum (SC). Methods: In this randomized clinical experiment of 30 Caucasian participants (15 with AD and 15 healthy controls), the superficial SC lipid profile was assessed through tape stripping non-lesional skin following treatment thrice daily for seven days with a moisturizer, and subsequently compared with untreated skin. Results: No discernible disparity in superficial SC lipid quantity was evident between the AD group and the control group. However, the SC lipid composition diverged significantly, with the AD group exhibiting diminished levels of long-chain EO CERs (*p* = 0.024) and elevated levels of short-chain C34 CERs (*p* = 0.025) compared to healthy skin. Moisturizer application significantly reduced the total SC lipids and all lipid subgroups in both groups. Within the AD group, a non-significant inclination towards an augmentation in EO CERs (*p* = 0.053) and reduction in C34 CERs (*p* = 0.073) was observed. Conclusion: The recent identification of distinctions in SC lipid composition between AD and healthy skin was substantiated by our findings. Topical moisturizer application, despite reducing overall total lipids, indicated a potential tendency towards a healthier lipid constitution in AD skin.

## 1. Introduction

The skin barrier resides in the outer epidermal layer, the stratum corneum (SC), and is defective in some skin diseases [1]. The SC consists of corneocytes surrounded by protein-bound ceramides that form the cornified cell envelope, providing a scaffold for extracellular lipids [2]. The extracellular lipids organized in the lipid lamella are composed of 50% ceramides (CER), 25% cholesterol, 15% free fatty acids (FFA) and small amounts of phospholipid [3]. The primary barrier is in the lipid lamella, whose composition and architecture are crucial for normal function [2]. In atopic dermatitis (AD), alterations in the extracellular lipid composition and organization have been described in both lesional and non-lesional skin [4,5,6]. Particularly, a shorter acyl-chain length or total length of carbon atoms of the CER in the lipid lamella are found in patients with atopic dermatitis (AD) [4,7] and this is mirrored by alterations in lipid organization, skin barrier function and disease severity [4].

Topical application of lipid rich moisturizers has the potential to restore the skin barrier in AD and prolong the time to eczema flare [8] and is also used to treat and prevent xerosis cutis in healthy individuals [9]. Non-physiological or physiological lipids may be used; the latter can penetrate the SC and be included in the endogenous pool of lipids, leading to a change in the barrier homeostasis [2]. In AD patients, the compositions of applied cholesterol, ceramides and fatty acids are of importance [2,10]. The effect of topical lipid rich moisturizer application to support healthy skin is not known. Previous studies suggested a preventive effect by early topical lipid rich moisturizer application in infants with a high risk of developing AD [11,12], but later studies found no such effect [13,14].

The aim of this study was to compare the SC lipid profile in non-lesional skin in patients with AD and controls (C) with and without short-term topical lipid rich moisturizer application.

## 2. Methods

This randomized clinical experiment was approved by the local ethics committee (H-18058392), the Data Protection Agency and conducted according to the Declaration of Helsinki.

### 2.1. Study Population and Intervention

In total, 30 Caucasian participants, 15 with AD and 15 age- and gender-matched controls, were included from September 2020 to June 2021. The AD patients had current dermatitis and were diagnosed by a physician using the UK Working Party Diagnostic criteria [15]. At baseline, blood samples were taken for filaggrin (FLG) genotyping, and treatment areas were randomized at four pre-defined (5 cm × 10 cm) areas on the upper inner arm and volar forearm, respectively (right/left allocation randomized). In the pertinent study, only two of these areas were included: one untreated area (T−) and one area treated thrice daily (T+) for seven days with 0.2 mL Doublebase Gel™ (Dermal Laboratories, Herts, UK) containing isopropyl myristate 15% *w*/*w* and liquid paraffin 15% *w*/*w*, glycerol, carbomer, sorbitan laurate, trolamine, phenoxyethanol and purified water. Participants were carefully instructed to wash hands prior to the application of moisturizer, apply it as a standardized procedure in the marked area and let the area air dry. The last moisturizer application was in the evening on day 7. At baseline and day 8, the pH was measured as a mean value of three measurements, and SC lipids were collected on day 8 by tape stripping of one treated and one untreated area.

Exclusion criteria were fragrance allergy, active infections, use of antibiotics, probiotics and fungicides within the past four weeks, pregnancy, breast feeding, and for controls, the criteria included a history of dermatitis. Within seven days and during the study period, participants were not permitted to use fragrance, treatment with topical corticosteroids, other topical immunosuppressants or moisturizers on/near exposure areas, and chlorinated pools, sauna and sun-exposure were forbidden. Further, fragrance-free shampoo and soap with a pH of 4.5–4.9 were provided to all participants to replace their own products during the study period. Showers were not permitted 12 h or less before the two study visits. All moisturizer containers were weighed during the first and last study visit.

### 2.2. Sampling and Analysis

Each treatment area was tape stripped once using fifteen consecutive tapes (3.8 cm^2^, D-Squame^®^; Monaderm, Monaco, France). Tapes were placed on the most central part of all test areas, and after use of a pressure applicator (225 g/cm^2^) (D-squame^®^; Monaderm, Monaco, France) for approximately 10 s, the tape was gently removed with a quick uniform movement with a plastic tweezer. Tape strip no. 3 was analyzed using Shotgun Lipidomics platform by Lipotype GmbH (Dresden, Germany), as described previously and in the subsequent section [16]. A total of 16 lipid classes were measured, which covered the main 12 ceramide subclasses, diacylglycerol, cholesteryl ester, triacylglycerol, and cholesterol [17].

Skin pH was measured with the Mettler-Toledo Seven2Go pH meter (mV meter S2 with a surface probe) (Sigma-Aldrich^®^, St. Louis, MO, USA) and the mean value of triplet measurements was used.

From blood samples, genomic DNA was purified and typed for the FLG loss-of functions mutations R501X, 2282del4 and R2447X [18].

### 2.3. Lipid Analysis

Methanol, propan-2-ol, chloroform, acetyl chloride and ammonium acetate were of analytical grade. Deuterated NS D3 (36:1;2) (cat# 2201) and EOS D9 (68:3;2) (cat# CUS9530) were purchased from Matreya LLC. Deuterated TAG D5 (cat# 110544) and DAG D5 (cat# 110538), and CE (20:0;0) (cat# 110870) were purchased from Avanti Polar Lipids.

Lipids were extracted using chloroform as described [16]. Samples were spiked with lipid class-specific internal standards prior to extraction. After drying and re-suspending in MS acquisition mixture, lipid extracts were subjected to mass spectrometric analysis. Mass spectra were acquired on a hybrid quadrupole/Orbitrap mass spectrometer equipped with an automated nano-flow electrospray ion source in both positive and negative ion modes. Lipid identification using LipotypeXplorer [19] was performed on unprocessed (*.raw format) mass spectra. For the MS-only mode, lipid identification was based on the molecular masses of the intact molecules. The MSMS mode included the collision-induced fragmentation of lipid molecules and lipid identification was based on both the intact masses and the masses of the fragments. Prior to normalization and further statistical analysis, lipid identifications were filtered according to the mass accuracy, occupation threshold, noise and background. Lists of identified lipids and their intensities were stored in a database optimized for the particular structure inherent to lipidomic datasets.

### 2.4. Statistical Analysis

Individuals were divided into groups according to their AD-status (AD vs. C) and the anatomical sites were divided into groups according to whether they were treated with moisturizers or not (T+ vs. T−). For the comparison of paired samples (T+ vs. T−), we used the paired sample t-test and for comparison of the unpaired samples (AD vs. C), we used the unpaired sample t-test. Lipid composition was analyzed in different ways. First, we calculated the total lipid amount where all 16 subclasses of lipids were pooled together. We then calculated the percent decrease of all the lipid classes after moisturizer treatment as compared to before. Also, we depicted the percent of the total amount of CERs according to the total length of the CERs evaluated by the number of carbon atoms. Furthermore, we calculated the average length of CERs. This was achieved using the following formula:$$\mathrm{Average}\;\mathrm{length}={\mathrm{length}}_{\mathrm{CER}}\times {\;\mathrm{fraction}}_{\mathrm{CER}}$$

In the above formula, length_CER_ was the total length of the CERs measured in carbon atoms and fraction_CER_ was the amount of the particular CER of the total. Finally, we calculated the relative abundance of ω-hydroxy fatty acid (EO) ceramides and the relative abundance of the 34-carbon atom length CERs.

Statistical analyses were performed using R version 3.6.0 (R Development Core Team 2018. R: A language and environment for statistical computing. R Foundation for Statistical Computing, Vienna, Austria).

## 3. Results

### 3.1. Clinical Characteristics

Clinical characteristics for the study are found in Table 1. The AD group had a median EASI of 6 (IQR: 2.6–19.3) corresponding to mild disease. AD patients had a higher pH in non-treated skin than the controls (5.74 vs. 5.21) (*p* = 0.0098). The pH did not change in any of the groups after moisturizer treatment (controls *p* = 0.328; AD *p* = 0.256). The actual usage of moisturizer compared to expected application was 80.0–81.8% in the controls and 80.1–80.6% in the AD group.

### 3.2. Lipids

Diacylglycerol and cholesterol esters were excluded from the analyses due to a large proportion of missing values caused by technical issues with the analyses. All other lipid classes were analyzed.

The total superficial SC lipid content did not differ significantly between AD and the controls (Figure 1). SC lipid composition differed at untreated sites, with the AD group having higher amounts of cholesterol (*p* = 0.013) and CER subclasses AS (*p* = 0.011), non-hydroxy-dehydrosphingosine (NdS) (*p* = 0.024), non-hydroxy-6-hydroxysphingosine NH (*p* = 0.01) and non-hydroxy-sphingosine (NS) (*p* = 0.003) than the controls, whereas no difference was seen in Triacylglycerol (TAG) or the CER subclasses Omegahydroxy-dehydrosphingosine (EOdS), Omegahydroxy-phytosphingosine (EOP), Alphahydroxy-dehydrosphingosine (AdS), Alphahydroxy-6-hydroxysphingosine (AH), Alphahydroxy-phytosphingosine (AP), Omegahydroxy-6-hydroxy-sphingosine (EOH), Omegahydroxy-sphingosine (EOS) and Non-hydroxy-phytosphingosine (NP).

Areas treated with moisturizer had significantly lower total superficial SC lipid contents compared with untreated areas for both AD and the controls (*p* < 0.001) (Figure 1). Initially, the SC matrix for the AD vs. the control group was composed of 42.9% vs. 26.1% cholesterol, 30.8% vs. 45.3% TAG and 26.3% vs. 28.6% CERs. The CER chain length distribution in all samples ranged from 34 to 58 and 63 to 78 (Appendix A). In the AD group, the EO CER subgroup (chain length 63–78 C atoms) was composed mainly of EOS (52.3%) and EOH (39.3%), and to a very low extent, EOdS (2.4%) and EOP (6.1%). Skin areas treated with moisturizer had significantly lower amounts of SC cholesterol, TAG and all 12 CER subclasses (Figure 2). Overall, the decrease in lipids after treatment was larger in the controls than the AD group. The range of the decrease was 36.0% to 71.1% for the controls and 16.2% to 62.0% for AD patients.

We found that untreated AD patients had a non-significant trend toward a lower average CER chain length compared with the controls (*p* = 0.056) (Figure 3A). Healthy controls had a higher relative amount of EO CERs compared with AD patients (*p* = 0.024) (Figure 3B). In both the AD and control groups, a non-statistically significant increase in the amount of EO CERs was found after treatment with moisturizer (*p* = 0.053 and *p* = 0.086, respectively) (Figure 3B). The amount of C34 CERs was higher in the AD group compared with the control group (*p* = 0.025) and in AD patients, a non-significant decreasing trend was found after moisturizer treatment (*p* = 0.073) (Figure 3C).

## 4. Discussion

In this study, we found alterations in the composition, but not in the quantity, of lipids in the superficial SC in clinically normal skin of AD patients compared with the controls. Further, we found that moisturizer treatment decreased the total amount of lipids in the superficial SC in both patients with AD and the controls. This finding was consistent for all lipid classes. In AD patients, areas treated with moisturizer had a non-significant higher amount of EO CERs and lower amount of C34 CERs.

The dysfunction of the skin barrier is evident in both lesional and non-lesional skin in AD, with factors such as mutations in the filaggrin gene (FLG) and variations in SC lipid composition and organization being of paramount importance [4,7,20]. The observed differences in SC lipid composition between the AD patients and the controls align with existing literature. Numerous studies have highlighted a reduction in ceramide chain length in AD patients [4,7,21], with a decrease in EO CERs and increase in C34 CERs, being more pronounced in lesional than non-lesional skin [7,22]. In our study, a non-significant elevation in the average ceramide chain length was noted in the control group compared with AD patients (*p* = 0.056). This was accompanied by a higher relative abundance of EO CERs (*p* = 0.024) and lower amounts of C34 CERs (*p* = 0.025). The non-significant nature of the average chain length difference in our study may be attributed to the inclusion of only non-lesional skin in a limited study population. Previous research by Janssen et. al. underscored the association between the low carbon chain length of CERs in non-lesional skin of AD patients and compromised SC lipid organization and decreased barrier function [4]. This suggests that the CER chain length is pivotal for proper lipid organization and skin barrier function in non-lesional AD skin [4]. Other studies have demonstrated a positive correlation between short-chain ceramides and transepidermal water loss (TEWL), while a negative correlation exists between long-chain ceramides and TEWL [7].

We found no difference in the total superficial SC lipid amount in non-lesional skin of patients with AD and the controls. This points toward the composition of SC lipids, and to a lesser extent, the total SC lipid amount being of importance for the decreased skin barrier in AD, which is in line with the existing literature [5,23]. Nevertheless, AD is a heterogenous condition, and the limited study population combined with individual differences among AD patients may conceal an actual difference.

Strategies to counteract AD are an area of ongoing extensive research. Moisturizers are commonly employed as the first-line treatment for xerosis and eczema flares [9,24]. Compromised skin barrier integrity precedes the manifestation of dermatitis [25]. Applied moisturizers have varied physiological effects. Traditional moisturizers, like petrolatum, is a non-physiological ingredient that may fill the extracellular spaces of the SC [2]. Conversely, physiological lipid precursors, such as those found in lamellar bodies, have the potential to permeate the SC and integrate with the endogenous lipid pool [2,26].

The moisturizer lipid compositions of cholesterol, ceramides and fatty acids have been shown to be of importance, and an excess or deficiency of a particular lipid may negatively affect lamellar body production formation [2]. Interestingly, in our study, moisturizer treatment resulted in a significant reduction in the total superficial SC lipid content and all measured lipid subclasses in both the control and AD group. In AD skin, moisturizer application induced a non-significant reduction in the relative amount of short-chain C34 CERs and a non-significant increase in the relative amount of long-chain EO CERs. This suggests that moisturizer application may induce a healthier SC lipid profile. The Doublebase Gel™ used in this study contains a combination of emollients (isopropyl myristate, liquid paraffin), humectants (glycerol), and occlusive agents. The decrease in SC lipid amount may be due to moisturizer-induced hydration, resulting in an increased water content filling the intercellular spaces between the corneocytes. This temporal dilution in the SC lipid concentration may have contributed to the observed decrease in total lipid content. Another speculation is that the physiological SC lipid production is suppressed by moisturizer application. More research in this area is needed.

The two main ingredients in our moisturizer (Doublebase Gel) were isopropyl myristate, which is composed of isopropyl alcohol and myristic acid, a common, naturally occurring fatty acid and liquid paraffin, which is a non-physiological component. Most studies report a beneficial effect of ceramide-containing formulations on dry skin and barrier function in patients with AD [2,27]. In a firm-sponsored study of 34 adults with dry skin, a moisturizer with physiological lipids showed superiority compared with one containing paraffin [10]. On the other hand, a large, randomized study found no difference in eczema severity in children with AD following 16 weeks of twice daily application of either lotion, cream, gel, or ointments [28]. It is reasonable to assume that another test moisturizer in a similar study set-up would result in a different result. Our test moisturizer was chosen, as it is identical to the one used in a previous study to explore whether regular moisturizer application can prevent AD. Chalmers et al. [14] randomized 1394 infants at high risk of developing AD to either daily use of moisturizers or no intervention for the first year of living and found no difference in eczema development at age 2 years. Another large, randomized population-based study of neonates found no preventive effect of 4-weekly oil baths containing paraffin liquid and trilaureth-4-phosphate following ceridal cream with non-physiological ingredients to the face at the age of 1 year [13].

The strengths of our study include the randomized controlled design with age- and gender-matched groups and standardized regimen with supplied shampoos and soaps, which limit the risk of detrimental skin barrier effects from irritants. Limitations include the relatively small study population, lack of normalization to protein amount on the tape, and the use of only one tape strip. In previous SC studies, tape strips no. 3 or 4 have mostly been used for lipid measurements [22,29,30], which eases comparisons with our results, although others have been using several tapes pooled together [31]. Further, SC lipids and especially the CERs, which were the main interest of this study, have been shown to have a limited variability according to depth [16]. In addition, protein-bound ceramides were not measured, which has been performed by others [32]. Another limitation is the lack of baseline lipid measurements. Although the upper and lower inner arm used as test areas in this study are two different anatomical locations, a previous study showed very similar CERs between the proximal lower forearm and the cubital fossa [5]. Free fatty acids were not determined in this study. Previous research has demonstrated the same pattern as for ceramides, with increased short-chain FFA and decreased long-chain FFA in both lesional and non-lesional AD skin compared with healthy controls [7].

In conclusion, our study highlights differences in superficial SC lipid matrix composition between clinically normal skin in AD and the controls. Moisturizer-treated skin had significantly lower total superficial SC lipids and all lipid subgroups in both patients with AD and the controls. The application of moisturizers entailed a non-significant trend towards a healthier lipid constitution, which was more pronounced in the AD group.

## Figures and Tables

**Figure 1 life-14-00345-f001:**
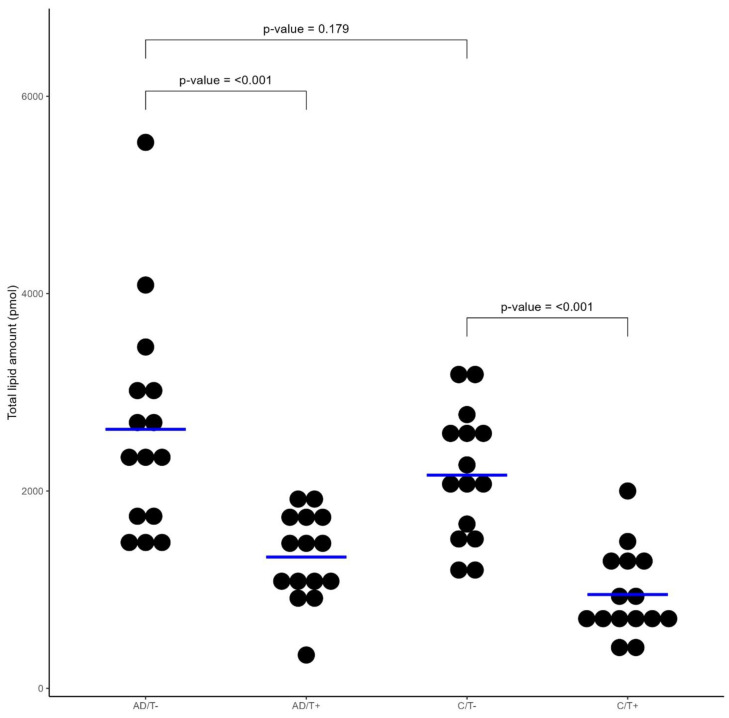
Total lipid amount in treated and untreated skin in AD and healthy controls. Dot plot visualizing the total lipid amount (pmol) for patients with atopic dermatitis (AD) and controls (C) with (T+) and without (T−) treatment with moisturizer. Blue line = mean. No difference was found at the non-treated site between AD and the controls, but a significant decrease was found at the treatment areas for both groups.

**Figure 2 life-14-00345-f002:**
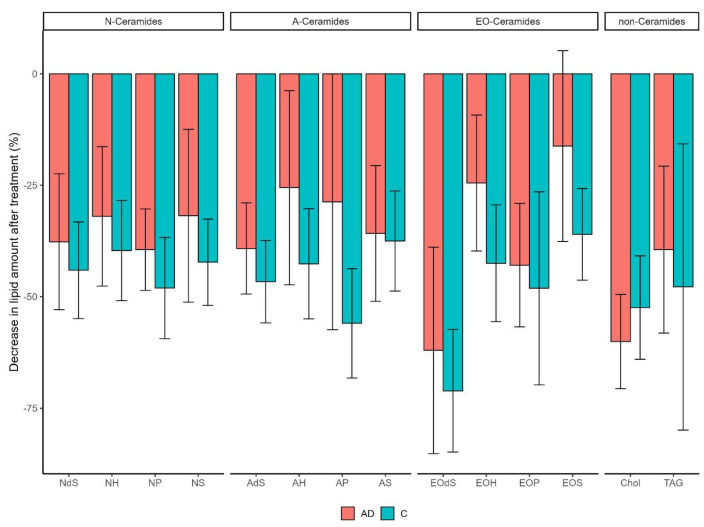
Changes in SC lipids after moisturizer treatment. Bar plot showing the decrease in the amount of lipid after moisturizing treatment divided by lipid class (upper graph) and subclass (*x*-axis) in patients with atopic dermatitis (AD) and controls (C). The bar plot shows the mean and the error bar shows the 95% CI of the mean.

**Figure 3 life-14-00345-f003:**
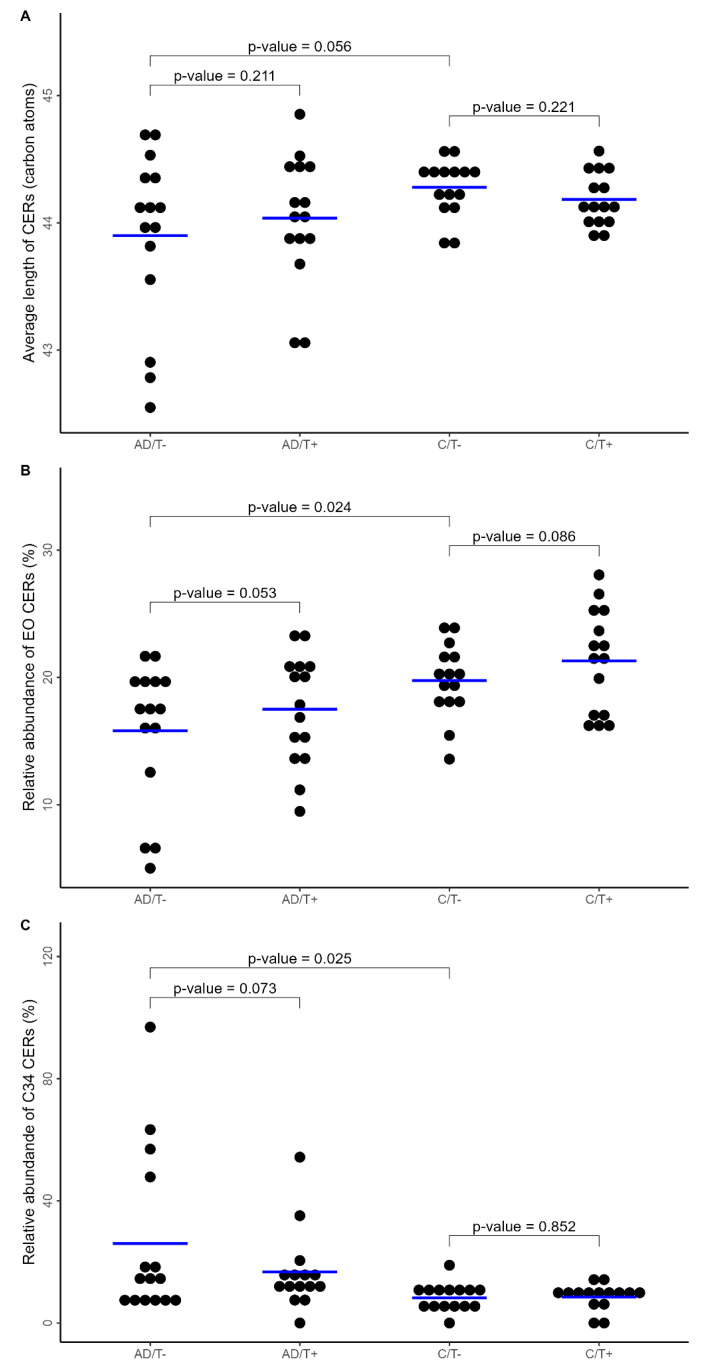
(**A**–**C**) Ceramides in untreated and treated skin of AD and healthy controls. Dot plots illustrating (**A**) the average CER chain length (carbon atoms) in patients with AD and controls with and without moisturizer treatment. In untreated skin (T−), there was a tendency to display a higher average CER chain length in controls than in AD patients (*p* = 0.056). There was no change after treatment (T+) in any of the groups. (**B**) The relative amount of long-chain ceramides (EO CERs) in the groups, illustrating a significantly higher amount in controls compared with AD patients (*p* = 0.024) and a non-significant increase with moisturizer treatment, more pronounced in the AD group (*p* = 0.053) than the control group (*p* = 0.086). (**C**) The relative amount of the short-chain ceramides with 34 carbon atoms, demonstrating a significantly lower amount in non-treated control skin compared with AD skin (*p* = 0.025) and a non-significant trend towards a lower amount in skin treated with moisturizer the AD-group (*p* = 0.073), but not in the control group (*p* = 0.852). Blue lines are illustrating the mean values.

**Table 1 life-14-00345-t001:** Clinical characteristics of the two study groups, with values in % (number) if not otherwise noted.

Value	AD Patients (n = 15)	Controls (n = 15)
Males	40% (6)	40% (6)
Age years (median (IQR))	25 (23–51.5)	25 (23–51.5)
FLG-mutation heterozygot	13% (2)	7% (1)
EASI (median (IQR))	6 (2.6–19.3)	-

## Data Availability

Data available on request due to restrictions.

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
