# Peer review of "Stratum Corneum Lipids in Non-Lesional Atopic and Healthy Skin following Moisturizer Application: A Randomized Clinical Experiment"

_life, 2024, doi:10.3390/life14030345_

Round 1

Reviewer 1 Report

Comments and Suggestions for Authors

The manuscript entitled: “Stratum corneum lipids in non-lesional atopic and healthy skin following moisturizer application: a randomized clinical experiment” is presenting some results obtained on the ceramides from stratum corneum of patients with atopic dermatitis.

Concerning the methods section 2.1 they could present a subchapter about the detailed composition of the Doublebase Gel. This moisturizing cream will have at least a moisturizing activity (

Short et al, Clinical and Exp. Dermatol, 2006, 10.1111/j.1365-2230.2006.02297.x) because it will help to recover the stratum corneum barrier by inducing the epidermal homeostasis and differentiation. To compare the control skin to the DA skin before and after treatment, they should compare them in the same conditions. That means that on the control skin it should be applied a placebo formula. 

Concerning the methods section 2., they should give the bibliographic reference for blood samples filaggrin genotyping.

Concerning the methods section 2.2, they show that they took 15 tape strips and only the nr.3 was analyzed. Is it enough to validate the results? Could they argue on this decision?

Some papers show that is necessary to consider the whole amount of the lipids from one spot (whole or at least half - one out of two consecutive tape strips) in order to have an appropriate amount of the lipid species (Popa et al., Arch Dermatol Res. 2011, 10.1007/s00403-011-1120-5; Reiter et al., Vet Dermatol, 2008, 10.1111/j.1365-3164.2009.00759.x; Imokawa et al, J Invest Dermatol, 1991, 10.1111/1523-1747.ep12470233) in order to see how the treatment will change the structure of the stratum corneum. There is a well-known heterogeneity in the stratum corneum from the surface to the deepest layer concerning the lipid composition and taking only one strip does not give a relevant distribution in the whole stratum corneum.

They should add the methods and the reagents that they use to extract the lipids from the strips, methods, reagents and solvents that they used for the whole lipids extraction and species lipids separation and the appropriate standards that helped to attest that they purified specific lipid fractions.  The paper (Sadowski, Sci.Rep, 2017, 10.1038/srep43761) that they quoted does not show the methodology of lipids extraction and species separation.

They assume that they measure a total of 16 lipid classes, which cover 12 ceramide subclasses, diacylglycerol, cholesteryl ester, triacylglycerol, and cholesterol. How were discarded the free fatty acids from the other purified lipids. Concerning the long-chain CERs species that are natively linked to proteins, they should give some information about their purification step (Lyuben et al, J Biol Chem, 1998, 10.1074/jbc.273.28.17763; Wertz et al, J Invest Dermatol,1989, 10.1111/1523-1747.ep13071317)

Concerning the methods section 2.3, they should give the reference for the used formula of calculation of the average length.

Concerning the lipids section 3.2 they emphasize that diacylglycerol and cholesteryl were excluded from the 16 lipids species but they do not give any information about the triglycerides and cholesterol ester besides the ceramides content.

It would be more representative if they calculate the total amount of the lipids species (eluted from 15 strips) and present in a table as total lipids/protein content and ceramides/ protein (ng/mg) for the AD patients before and after treatment versus the control.

Somehow, the ceramide content (presented in figures 3) taken from one strip is not relevant for the whole spot taken in 15 strips successively from the outmost layer to the deepest layer nearing the granulosum layer of the stratum corneum. 

For the sentence “We then calculated the percent decrease of all lipid classes after moisturizer treatment as compared to before”, they should give some clarification, because usually the lipids amount is restored after an AD treatment. In this respect it was suggested by Palmer C.N .et al. (Palmer et al, Nat. Genet., 2006,10.1038/ng1767) that in AD patients the loss-of-function mutations for keratinocyte-differentiation related-genes encoding filaggrin could be an important risk factor.

Nevertheless, the paper could be considered, provided that methods and results are more properly presented.

Author Response

Reviewer 1

The manuscript entitled: “Stratum corneum lipids in non-lesional atopic and healthy skin following moisturizer application: a randomized clinical experiment” is presenting some results obtained on the ceramides from stratum corneum of patients with atopic dermatitis.

1.ANSWER: First, we would like to thank the Editor and the reviewer for their time and comments upon the manuscript. We have updated the manuscript with the reviewer’s comments. Please find below a point-by-point answer

Concerning the methods section 2.1 they could present a subchapter about the detailed composition of the Doublebase Gel. This moisturizing cream will have at least a moisturizing activity (Short et al, Clinical and Exp. Dermatol, 2006, 10.1111/j.1365-2230.2006.02297.x) because it will help to recover the stratum corneum barrier by inducing the epidermal homeostasis and differentiation. To compare the control skin to the DA skin before and after treatment, they should compare them in the same conditions. That means that on the control skin it should be applied a placebo formula. 

 2.ANSWER: Thank you for this comment and request. The full ingredient list for Doublebase Gel has been added to the method section. In the discussion section, the mechanism of the active ingredients of the Double base gel is discussed, and also the general difference in the action of moisturizers is touched upon, e.g. physiological verses non-physiological ingredients. Regarding the comment on comparing the skin with moisturizer with a placebo formula, this was not the aim. The aim was to compare the SC lipid profile in non-lesional skin in patients with AD and controls, with and without short-term topical lipid rich moisturizer application.

Concerning the methods section 2., they should give the bibliographic reference for blood samples filaggrin genotyping.

  1. ANSWER: The bibliographic reference is already found in the end of sections 2.2 (reference 18).

Concerning the methods section 2.2, they show that they took 15 tape strips and only the nr.3 was analyzed. Is it enough to validate the results? Could they argue on this decision? Some papers show that is necessary to consider the whole amount of the lipids from one spot (whole or at least half - one out of two consecutive tape strips) in order to have an appropriate amount of the lipid species (Popa et al., Arch Dermatol Res. 2011, 10.1007/s00403-011-1120-5; Reiter et al., Vet Dermatol, 2008, 10.1111/j.1365-3164.2009.00759.x; Imokawa et al, J Invest Dermatol, 1991, 10.1111/1523-1747.ep12470233) in order to see how the treatment will change the structure of the stratum corneum. There is a well-known heterogeneity in the stratum corneum from the surface to the deepest layer concerning the lipid composition and taking only one strip does not give a relevant distribution in the whole stratum corneum.

  1. ANSWER: Thank you for this comment. The mentioned studies are 2 studies of SC lipids in dogs using cyanoacrylate stripping method to obtain the SC lipids in dogs (Reiter et al and Popa et al), and one study of SC collection in humans, (Imokawa et al) also stripped with cyanoacrylate resin. The latter study is demonstrating that not all of SC is collected, which is also of course the case with our collection method (see Sølberg et al. Skin tape stripping: Which layers of the epidermis are removed? Contact Dermatitis. 2019;80(5):319-321. doi:10.1111/cod.13199) There are several recent studies in humans in this area of research only using one tape strip for lipid SC analyses (see strengths/limitations of the discussion section), and tape 3 or 4 are usually chosen in these studies (eg. Kezic 2022 BJD ref 28/Andersson 2023 JEADV ref nr. 21/Rinnov 2022 Allergy ref nr 27/Tonic 2020 IJMS ref nr 6). As is also mentioned in this section, the SC lipids, and especially the ceramides which were the main interest in this study, have been shown to have limited variability with depth (Ref Sadowski 2017 ref. 16). However, using one strip is mentioned as a limitation of this study.

They should add the methods and the reagents that they use to extract the lipids from the strips, methods, reagents and solvents that they used for the whole lipids extraction and species lipids separation and the appropriate standards that helped to attest that they purified specific lipid fractions.  The paper (Sadowski, Sci.Rep, 2017, 10.1038/srep43761) that they quoted does not show the methodology of lipids extraction and species separation.

  1. ANSWER: See information added as supplementary.

They assume that they measure a total of 16 lipid classes, which cover 12 ceramide subclasses, diacylglycerol, cholesteryl ester, triacylglycerol, and cholesterol. How were discarded the free fatty acids from the other purified lipids. Concerning the long-chain CERs species that are natively linked to proteins, they should give some information about their purification step (Lyuben et al, J Biol Chem, 1998, 10.1074/jbc.273.28.17763; Wertz et al, J Invest Dermatol,1989, 10.1111/1523-1747.ep13071317)

6.ANSWER: – Lipotype shotgun lipidomics is based on direct infusion mass spectrometry (MS) without prior chromatographic separation. This MS-method is combined with proprietary workflow of lipid identification and quantification, which is adjusted for lipids within specific mass range and enables broadest coverage for stratum corneum lipids. Free fatty acids are not within mass range of this particular method, therefore this platform technology allows to identify and quantify only complex lipids, particularly 12 sub-classes of ceramides, DAG, TAG, cholesterol esters, as well as free cholesterol. Protein-bound ceramides are not quantified with this approach, as ceramides are bound to proteins covalently; covalent bonds are not disrupted during lipid extraction used within this method.

Concerning the methods section 2.3, they should give the reference for the used formula of calculation of the average length.

7.ANSWER: Thank you for this request. A reference has been added. 

Concerning the lipids section 3.2 they emphasize that diacylglycerol and cholesteryl were excluded from the 16 lipids species but they do not give any information about the triglycerides and cholesterol ester besides the ceramides content.

  1. ANSWER: Thank you very much for noticing an error in the manuscript. It is the cholesterol ester, together with the mentioned diacylglycerol that were excluded, due to a large proportion of missing values. This has been corrected. Results on triglycerides are available (see Figure 2 and the 2nd section in 3.2).

It would be more representative if they calculate the total amount of the lipids species (eluted from 15 strips) and present in a table as total lipids/protein content and ceramides/ protein (ng/mg) for the AD patients before and after treatment versus the control.

  1. ANSWER: Thank you for this suggestion. Unfortunately, these calculations cannot be done, as lipids were not extracted from all tapes, and the protein content was not measured.

Somehow, the ceramide content (presented in figures 3) taken from one strip is not relevant for the whole spot taken in 15 strips successively from the outmost layer to the deepest layer nearing the granulosum layer of the stratum corneum. 

  1. ANSWER: See our previous answer.

For the sentence “We then calculated the percent decrease of all lipid classes after moisturizer treatment as compared to before”, they should give some clarification, because usually the lipids amount is restored after an AD treatment. In this respect it was suggested by Palmer C.N .et al. (Palmer et al, Nat. Genet., 2006,10.1038/ng1767) that in AD patients the loss-of-function mutations for keratinocyte-differentiation related-genes encoding filaggrin could be an important risk factor.

  1. ANSWER: Thank you for this question. In this clinical randomized trial, we found a decrease of total lipid amount and all lipid classes following moisturizer treatment. Previously well-known alterations in SC lipid composition between AD and control skin was also found. These alterations are shown in non-lesional skin. The understanding of the SC lipid changes, and the underlying mechanisms are not well understood. In the discussion section, it is speculated upon, whether the physiological SC lipid production is suppressed by moisturizer application, thereby leading to decreased amounts.

Nevertheless, the paper could be considered, provided that methods and results are more properly presented.

12.ANSWER: We hope the answers and revisions are fulfilling the requests. 

Reviewer 2 Report

Comments and Suggestions for Authors

The manuscript „Stratum corneum lipids in non-lesional atopic and healthy 2 skin following moisturizer application: a randomized clinical 3 experiment“ represents a very interesting and well performed study. However, some minor things should be corrected before publication, as follows:

1.      Please in part 2.2. explain how the pressure applicator was used, why tape no.3 was measured and using what metholdology.

2.      How the authors explain the fact that the total SC lipid content did not differ significantly between AD and controls? However, the SC lipid composition differed.

3.      It is also worth explaining that areas treated with the moisturizer had significantly lower total SC lipid content compared with untreated areas for both AD and controls. Please explain this more.

4.      Please indicate the whole composition of the used moisturizer aa well as an explaination how this moisturizer was chosen. On one place it is mentioned that this preparation of moisturizers has been previously used. However, we do not know the whole composition in the article. Usually in atopic dermatitis also plant oils are used, such as Oenothera biennis oil.

5.      The error bars in Fig.2 are so high, that there should be no difference between AD and control skin. Please check again the significance, a nd add the full name for AD and C in the figure caption.

6.      In general, please expand the discussion part, adding more explanation to the obtained results.

Author Response

Reviewer 2

The manuscript „Stratum corneum lipids in non-lesional atopic and healthy 2 skin following moisturizer application: a randomized clinical 3 experiment“ represents a very interesting and well performed study. However, some minor things should be corrected before publication, as follows:

ANSWER: Thank you very much for your kind words.

Please in part 2.2. explain how the pressure applicator was used, why tape no.3 was measured and using what metholdology.

  1. ANSWER: Thank you for this request. The use of the pressure applicator is described in depth. The methodology of the lipid extraction and measurements are described in the reference (16), further a supplementary section has been added to fulfil this request. In the discussion section, the choice of tape strip number 3 for lipid measurements are justified. In short, in previous studies of SC lipids in AD and control skin, tape strip number 3 og 4 are usually used (eg. Kezic 2022 BJD ref 28/Andersson 2023 JEADV ref nr. 21/Rinnov 2022 Allergy ref nr 27/Tonic 2020 IJMS ref nr 6), also the SC lipids, and especially the ceramides which were the main interest in this study, have been shown to have limited variability with depth(Ref 16, Sadowski) .

How the authors explain the fact that the total SC lipid content did not differ significantly between AD and controls? However, the SC lipid composition differed.

  1. ANSWER: Thank you for this interesting question. Previous studies have demonstrated altered SC lipid composition in non-lesional AD skin compared with control skin (Janssens et al) and several studies have shown these differences in lesional skin. The same alterations, with lower amounts of long-chain ceramides, and a higher amount of short-chain ceramides were also shown in the present study. It is speculated, that the actual lipid amount is of less importance, and that the composition is the main player. This speculation has been added to the discussion section.

It is also worth explaining that areas treated with the moisturizer had significantly lower total SC lipid content compared with untreated areas for both AD and controls. Please explain this more.

  1. ANSWER: You are right. A couple of lines have been added to the discussion regarding this issue. From the results obtained, the reason for this finding can only be speculated upon. In short, application of this moisturizer seems to lead to a more balanced lipid profile, although, as mentioned in the discussion the extent and specific changes of moisturizers in general, vary depending on the formulation and ingredients of the moisturizer. As the physiological ingredients in this formulation may penetrate the SC and integrate with the lipid pool, a possible suppression of natural production may be the result and explain the findings.

Please indicate the whole composition of the used moisturizer aa well as an explaination how this moisturizer was chosen. On one place it is mentioned that this preparation of moisturizers has been previously used. However, we do not know the whole composition in the article. Usually in atopic dermatitis also plant oils are used, such as Oenothera biennis oil.

  1. ANSWER: Thank you for the suggestion. The whole composition of Doublebase gel has been added to the method section.

The error bars in Fig.2 are so high, that there should be no difference between AD and control skin. Please check again the significance, a nd add the full name for AD and C in the figure caption.

  1. ANSWER: Figure 2 shows the decrease in the amount of each studied lipid following moisturizer treatment, and no differences between AD and control skin are to be deduced from this figure. The full names for AD and C have been spelled out in the figure caption.

In general, please expand the discussion part, adding more explanation to the obtained results.

  1. ANSWER: The discussion section has been expanded, concerning the previous comments on the results. We hope it is satisfactory.

Reviewer 3 Report

Comments and Suggestions for Authors

The manuscript presents a study carried out on patients with atopic dermatitis compared to healthy subjects, following the influence that treatment with emollients can have on the lipid composition and quantity of the epithelial stratum corneum.

Both the strengths of the study and the weaknesses are presented, highlighting the lack of onset determinants. In addition, I believe that a description of the stage of atopic dermatitis at onset would have been useful, with a description of the affected areas. Since the treatment was administered on certain anatomical regions, it was important to know whether or not those regions were affected by the disease. Possibly, the authors could complete with some aspects in this regard.

Author Response

Reviewer 3

The manuscript presents a study carried out on patients with atopic dermatitis compared to healthy subjects, following the influence that treatment with emollients can have on the lipid composition and quantity of the epithelial stratum corneum.

Both the strengths of the study and the weaknesses are presented, highlighting the lack of onset determinants. In addition, I believe that a description of the stage of atopic dermatitis at onset would have been useful, with a description of the affected areas. Since the treatment was administered on certain anatomical regions, it was important to know whether or not those regions were affected by the disease. Possibly, the authors could complete with some aspects in this regard.

ANSWER: Thank you for your request. The EASI score of 6 is found in section 3.1, corresponding to mild disease. The tested regions in the pertinent study were non-lesional.

Round 2

Reviewer 1 Report

Comments and Suggestions for Authors

The manuscript entitled: “Stratum corneum lipids in non-lesional atopic and healthy skin following moisturizer application: a randomized clinical experiment” is presenting some results obtained on lipids and ceramides from stratum corneum of patients with atopic dermatitis extracted from only one strip (number 3) out of 15.

The basic issue is that the quality of obtained results is not enough representative and robust because they based their results only on a single strip evaluation which is not representative for the whole Stratum corneum structure and composition.   

Many research articles showed before that the Stratum Corneum is has a similar in lipids and protein content and chemical compostion but the relative proportions differ from the top to the bottom of the Stratum Corneum sublayers (from Stratum Corneum Compactum to Stratum Corneum Disjunctum) in healthy skin. This is due to the keratinization metabolism in mammalian skin (already showed in human, dogs and mice skin). This heterogeneity is more obvious in atopic skin as it was reported in many research articles (example of electron microscopy of the Stratum Corneum- Inman et all, 2001).

Moreover, basic papers in the field showed the sex-dependency of ceramides in the SC of people (Denda et all, 1993) and that the ceramide content of the SC significantly declines with increasing age in people (Yamamura T et all, 1990) and an acidification of the SC is taking place in aged skin (WangZ et all, 2020).

Based on these facts, the authors should consider the extraction of lipids from whole strips and the comparison of the lipid content of inner strips with outer strips. This will give an authentic comparison for the lipids.

Regarding the purification of protein-bound ceramides, the adequate protocol is more complex and it is described first (Lyuben et all, 1988). It states that the aliquots of Epidermal Cornified Cell Envelope containing protein-bound ceramides should be resuspended in 1 M KOH in 95% methanol for 15–120 min at 45 °C, washed with methanol and dried, before the released ceramides can be taken up in chloroform/methanol (95:5 V/V).

 Based on these observations, the obtained results are clearly not a reflection of the actual composition in whole Stratum Corneum lipids and protein-bound ceramides, and the paper is not acceptable as such at this stage.

Yamamura T, Tezuka T. Change in sphingomyelinase activity in human epidermis during aging. J. Dermatol. Sci. 1990; 1: 79–84. DOI : 10.1016/0923-1811(90)90219-4

Denda M, Koyama J, Hori J. et al. Age- and sex-dependent changes in stratum corneum sphingolipids. Arch. Dermatol. Res. 1993; 285: 415–7. DOI: 10.1007/BF00372135

Wang Z, Man MQ, Li T, Elias PM, Mauro TM. Aging-associated alterations in epidermal function and their clinical significance. Aging (Albany NY). 2020 Mar 27;12(6):5551-5565. DOI: 10.18632/aging.102946. Epub 2020 Mar 27.

Lyuben N., Marekov and Peter M. Steinert, Ceramides Are Bound to Structural Proteins of the Human Foreskin Epidermal Cornified Cell Envelope, J. Biol. Chem. Vol. 273, No. 28, Issue of July 10, pp. 17763–17770, 1998 , DOI: 10.1074/jbc.273.28.17763

Inman AO, Olivry T, Dunston SM, Monteiro-Riviere NA, Gatto H. Electron Microscopic Observations of Stratum Corneum Intercellular Lipids in Normal and Atopic Dogs. Vet. Pathol. 2001;38(6):720-723. doi:10.1354/vp.38-6-720

Imokawa G, Abe A, Jin K, Higaki Y, Kawashima M, Hidano A. Decreased level of ceramides in stratum corneum of atopic dermatitis: an etiologic factor in atopic dry skin? J. Invest. Dermatol. 1991 Apr;96(4):523-6. doi: 10.1111/1523-1747.ep12470233

Author Response

We agree with the reviewer that it would be preferable to have all tape strips analyzed, however due to economical and logistic considerations, additional analyzing of tape strips is not possible. Also, we do believe that our estimates are mirroring the physiological changes in the stratum corneum. The method we have used and the number of tape trips analyzed is standard of practice is many recent papers from experts in this field (eg. Kezic 2022 BJD ref 28/Andersson 2023 JEADV ref nr. 21/Rinnov 2022 Allergy ref nr 27/Tonic 2020 IJMS ref nr 6), as have been referred to in the discussion section, and in our previous response to the reviewer comments. Also, the SC lipids, and especially the ceramides which were the main interest in this study, have been shown to have limited variability with depth (Ref Sadowski 2017 ref. 16).

Sincerely, 

The Copenhagen Group

Round 3

Reviewer 1 Report

Comments and Suggestions for Authors

The obvious aim of this manuscript was to emphasize the interest of using mass spectrometry to assess the lipid composition of human stratum corneum from atopic and healthy subjects. However, there are two main drawbacks in this study.

The first and major one is the use of a single strip for the analysis. The results are inconsistent due to the obvious heterogeneity in the lipids of atopic stratum corneum with respect to the depth of the layers taken from the top to the bottom (15) by stripping and assessing only one.  (see my comment for the second review).

Several papers showed that is a difference between lipids content in atopic versus normal Stratum Corneum and more over that is a difference between the strips taken from the bottom or the top of the Stratum Corneum.

For example, the group of Bouwstra (Danso, 2017) was taking at least 4 consecutives strips (6 to 9 the deepest ones) out of 10 on atopic human Stratum Corneum to assess rigorously the lipids in order to have a robust analysis. Also, they (Boiten, 2016) mention that in order to quantify the amount of ceramide in multiple subjects using the tape-stripping procedure, it is important to distinguish biological variation and the variation caused by the tape-stripping procedure.

The same conclusion was drawn with the analyses of the 12 consecutive strips from the stratum corneum of atopic dogs (Popa et al, Arch. Dermatol. Res. 2011, 303 :433-440) and shown by HPTLC the heterogeneity in lipid classes of the strips from top to bottom.

With such an heterogeneity in biological variation from top to bottom of the Stratum Corneum of atopic stratum corneum, is difficult to draw conclusions from the lipid analysis of a single strip out of 10 or 15.

The second drawback of the present manuscript is the total lack of investigation on protein-bound ceramides in the stratum corneum that are of utmost importance for the efficacy of the skin barrier.

As it was first described (Marekov et all, 1988) the purification of protein-bound ceramides is more complex and needs adequate protocols for extraction. It states that the aliquots of Epidermal Cornified Cell Envelope containing protein-bound ceramides should be resuspended in 1 M KOH in 95% methanol for 15–20 min at 45 °C, washed with methanol and dried, before the released ceramides can be taken up in chloroform/methanol (95:5 v/v).

In the paper from Bouwstra lab (Boiten, 2016), in order to have a efficient lipid extraction the samples were saponified with 1M KOH in 90% MeOH for 1 hour at 60°C neutralized with 1M HCl.

In another paper (Popa et al, 2012), in order to release from the strips the protein-bound lipids, the cell pellet residues left in the glass tubes were treated by a mild saponification with 3 mL KOH 0.1 N in methanol: water 10 : 1 (v ⁄ v) for 2 h at 50 °C.

With the method presently used by the authors for lipid extraction, they completely ignore the presence of protein-bound ceramides that require such an alkaline methanolysis of the sample to be released (Marekov et al, 1988; Popa et al, 2012; Boiten et al, 2016).

Although the MS analysis carried out by the authors is at the state of the art level, the overall interest of this paper is quite questionable without additional studies taking in account the two points raised by the reviewer.

Regarding the issue related to the depth of the analyzed strip in the stratum corneum, an alternative possibility would be for the authors to carry out the lipid analysis on the pool of all strips. In this case the comparison between the data obtained with atopic and healthy stratum corneum would be of interest.

L. N. Marekov and Peter M. Steinert, Ceramides Are Bound to Structural Proteins of the Human Foreskin Epidermal Cornified Cell Envelope, J. Biol. Chem. Vol. 273, No. 28, Issue of July 10, pp. 17763–17770, 1998 , DOI:  10.1074/jbc.273.28.17763

W. Boiten, S Absalah, Vreeken R, Bouwstra J, J. van Smeden, Quantitative analysis of ceramides using a novel lipidomics approach with three dimensional response modelling , BBA - Molecular and Cell Biology of Lipids , 11 July 2016 , DOI: 10.1016/j.bbalip.2016.07.004

 M.O. Danso, W Boiten, V. van Drongelen, K.M. Gmelig, G. Gooris, A. El Ghalbzouri, S Absalah, Vreeken R, J. van Smeden, Bouwstra J, Altered expression of epidermal lipid bio-synthesis enzymes in atopic dermatitis skin is accompanied by changes in stratum corneum lipid composition, J Dermatol Sci (2017), DOI: 10.1016/j.jdermsci.2017.05.005

I. Popa, N. Remoue, B. Osta, D. Pin, H. Gatto, M. Haftek and J. Portoukalian , The lipid alterations in the stratum corneum of dogs with atopic dermatitis are alleviated by topical application of a sphingolipid-containing emulsion,  Clinical and Experimental Dermatology , 2011, DOI:10.1111/j.1365-2230.2011.04313.x

Author Response

Dear reviewer,

Thank you for your thorough work to improve our paper. As mentioned, the aim of the study was to investigate the superficial SC lipid profile in healthy and AD skin, but also compare possible changes after controlled moisturizer treatment. The last aim has not been studied exhaustedly by independent research groups, not sponsored by the industry.

Unfortunately, no more lipid analyses can be performed at this stage, as mentioned, due to logistic and economic reasons.

The manuscript has been revised to underscore that only the superficial SC was used. Also, according to your two draw-back points, both these weaknesses of the study were already included in the end of the discussion sections. However, suggested additional studies have been added. We can probably agree on, that analyzing lipids is very difficult, and not all analysis one wish for, can be included, as the studies you mention are lacking other measurements, and one is performed on dogs. However, we believe, that this study adds to the existing literature in this field, and provides new knowledge.